# Local Health Department COVID-19 Vaccination Efforts and Associated Outcomes: Evidence from Jefferson County, Kentucky

**DOI:** 10.3390/vaccines13090901

**Published:** 2025-08-26

**Authors:** Shaminul H. Shakib, Seyed M. Karimi, J. Daniel McGeeney, Md Yasin Ali Parh, Hamid Zarei, Yuting Chen, Ben Goldman, Dana Novario, Michael Schurfranz, Ciara A. Warren, Demetra Antimisiaris, Bert B. Little, W. Paul McKinney, Angela J. Graham

**Affiliations:** 1Department of Health Management and Systems Sciences, School of Public Health and Information Sciences, University of Louisville, Louisville, KY 40202, USA; hamid.zarei@louisville.edu (H.Z.); demetra.antimisiaris@louisville.edu (D.A.); bert.little@louisville.edu (B.B.L.); 2Eastern Research Group Inc., Concord, MA 01742, USA; daniel.mcgeeney@erg.com; 3Department of Bioinformatics and Biostatistics, School of Public Health and Information Sciences, University of Louisville, Louisville, KY 40202, USA; mdyasinali.parh@louisville.edu; 4Center for Health Equity, Louisville Metro Department of Health & Wellness, Louisville, KY 40202, USA; yuting.chen@louisvilleky.gov (Y.C.); benjamin.goldman@louisvilleky.gov (B.G.); dana.novario@louisvilleky.gov (D.N.); michael.schurfranz@louisvilleky.gov (M.S.); ciara.warren@louisvilleky.gov (C.A.W.); angela.graham@louisvilleky.gov (A.J.G.); 5Department of Health Promotion and Behavioral Sciences, School of Public Health and Information Sciences, University of Louisville, Louisville, KY 40202, USA; wpaul.mckinney@gmail.com

**Keywords:** COVID-19 vaccination, COVID-19 outcomes, cases, hospitalization, deaths, vaccine effectiveness, updated vaccines, long COVID, racial disparity, equity

## Abstract

**Background**: While disparities in vaccine uptake have been well documented, few studies have evaluated the impact of local vaccine programs on COVID-19 outcomes, namely cases, hospitalizations, and deaths. **Objectives**: Evaluate the impact of COVID-19 vaccine doses coordinated by the Louisville Metro Department of Public Health and Wellness (LMPHW) on COVID-19 outcomes by race across ZIP codes from December 2020 to May 2022 in Jefferson County, Kentucky. **Methods**: Fixed-effects longitudinal models with ZIP codes as ecological time-series units were estimated to measure the association between COVID-19 vaccine doses and outcomes with time lags of one week, two weeks, three weeks, four weeks, and one month. Models were adjusted for time (week or month of the year) and its interaction with ZIP code. **Results**: In the one-week lag model, significant negative associations were observed between LMPHW-coordinated vaccine doses and COVID-19 outcomes, indicating reductions of 11.6 cases, 0.4 hospitalizations, and 0.3 deaths per 100 doses administered. Vaccine doses were consistently associated with fewer deaths among White residents across all lags, with an average reduction of 0.2 deaths per 100 doses. No significant associations were found for Black residents. Temporal trends also indicated declines in COVID-19 outcomes when LMPHW’s vaccine administration program peaked, between March and May 2021. **Conclusions**: Timely uptake of COVID-19 vaccines remains critical in avoiding severe outcomes, especially with emerging variants. Racial disparities in vaccine–outcome associations emphasize the potential need for equitable, community-driven vaccine campaigns to improve population health outcomes.

## 1. Introduction

The COVID-19 pandemic has entered an endemic phase, yet transmission remains widespread. Between 8 and 14 June 2025, approximately 3.0% of the U.S. population—around 10.2 million individuals—tested positive for COVID-19 [1]. Continued transmission places individuals—particularly racial and ethnic minorities and COVID-19 survivors—at increased risk of infection, reinfection, and post-acute sequelae of SARS-CoV-2 infection (PASC), also known as “long COVID” [2,3,4].

Approximately 10% of individuals infected with COVID-19 develop long COVID [5]. Long COVID can affect multiple physiological systems that cater to immunity, cardiovascular health, and neurological and cognitive functioning [5]. The risk of developing long COVID is higher among Black or Hispanic survivors, as well as those who required hospitalization or intensive care during the acute stage of illness [2,3,4].

COVID-19 vaccination significantly reduces the risk of reinfection, hospitalization, in-hospital death, and long COVID [6,7,8,9,10,11]. However, its protective effect can diminish over time, particularly against newly emerging variants—a phenomenon known as waning immunity [6].

Beginning in 2023, strain-specific COVID-19 vaccine formulations were introduced to target circulating SARS-CoV-2 variants, with updated versions in 2024 and 2025 designed to enhance protection against the most prevalent strains. In contrast, earlier vaccine doses—referred to as “boosters”—relied on the same formulations to boost immunity, offering conventional protection against a broad range of virus strains. Updated vaccines have demonstrated effective protection against specific variants, namely JN.1 and XBB, which were primarily in circulation from September 2023 to January 2024 [7,8]. Therefore, the continuation of the COVID-19 vaccination campaign remains a critical public health priority to sustain vaccination efforts [9].

State and local public health agencies, in alignment with the Centers for Disease Control and Prevention (CDC), recognize persistent disparities in COVID-19 vaccine uptake by race and ethnicity and have committed to advancing vaccine equity [10]. Numerous studies have reported on racial and ethnic differences in vaccine uptake across geographic units—such as counties and ZIP codes—highlighting inequities, particularly among Black and other minority populations [11,12,13,14]. These findings help state and local public health agencies to make vaccination programs more equitable by directing resources to identified underserved populations.

Although disparities in COVID-19 vaccine uptake among racial groups have been well documented [11,12,13,14], few studies have assessed the impact of state and local vaccine programs on COVID-19 outcomes, namely, cases, hospitalizations, and deaths. Existing research has been primarily conducted at the state or regional level, with a focus on overall vaccine uptake and associated outcomes [15,16].

ZIP codes are relatively small geographic units that can capture group-level patterns in demographics and socioeconomic status (SES). Although individual-level variation may exist within a given ZIP code, access to resources—such as schools, housing, healthcare facilities, grocery stores, parks, recreational facilities, and gyms—as well as key SES indicators, including health insurance status, income, and education, is generally similar for residents within and across adjacent ZIP codes.

As a result, individuals living in proximate ZIP codes often experience similar health outcomes [17,18]. Consequently, if access to these resources and SES factors remain stable over time, including ZIP codes in analytic models can help account for time-invariant confounding effects, making them a reasonable, though limited, proxy for individual SES.

Assessing the impact of vaccine programs on COVID-19 outcomes—stratified by race—at a finer geographic level (i.e., ZIP codes) can reveal patterns and disparities that may be obscured at larger spatial scales. Such findings can help state and local public health agencies evaluate program effectiveness and inform strategies to reduce vaccine inequities.

This study estimates the time-lagged effects of COVID-19 vaccines coordinated by the Louisville Metro Public Health and Wellness Department (LMPHW) on outcomes (cases, hospitalizations, and deaths) within racial groups across ZIP codes in Jefferson County, Kentucky. The findings aim to inform the design and continuation of equitable vaccination campaigns.

## 2. Materials and Methods

### 2.1. Study Design

Fixed-effect longitudinal models, with ZIP codes as ecological time-series units and week or month as time units, were employed in this study. The study focused on Jefferson County, Kentucky, from December 2020 to May 2022 and included 38 ZIP codes with residential population. In 2021, Jefferson County had an estimated total population of 817,446, of which 520,643 (63.7%) identified as White, 171,549 (21.0%) as Black, and 125,254 (15.3%) as members of other racial groups [19]. This study was approved by the University of Louisville (IRB #: 22.0681) and the Kentucky Cabinet for Health and Family Services (IRB #: CHFS-IRB-DPH-FY22-33), and adhered to the Strengthening the Reporting of Observational Studies in Epidemiology (STROBE) guidelines [20].

### 2.2. Data Sources

#### 2.2.1. LMPHW Event Data

LMPHW is responsible for the public health services of Jefferson County, Kentucky. Its COVID-19 vaccine campaign initially coordinated doses through selective drive-through mass vaccination standing sites across the county. To support timely and equitable vaccine administration, LMPHW also deployed mobile units to reach individuals who were unable or unlikely to access these sites.

The mobile units primarily served homeless shelters and encampments, correctional facilities, substance use disorder treatment centers, domestic violence shelters, and communities facing cultural, language, technological, and transportation barriers. Licensed nurses administered the vaccines, with LMPHW staff overseeing operations.

In addition, dedicated mobile units were available upon request for homebound individuals—primarily older adults or those with disabilities—who could not travel to standing or mobile unit sites. LMPHW maintained a database that tracked the daily number of vaccine doses delivered through each modality by ZIP code. Vaccine counts from all modalities were combined into a single variable to estimate the overall impact of LMPHW’s coordinated vaccination efforts, irrespective of delivery setting.

#### 2.2.2. Kentucky State Contact Tracing and Tracking (CTT)

The CTT database, maintained by local public health departments, contains daily records of individual-level COVID-19 cases along with supplementary information, including race, residential ZIP code (i.e., home address), hospitalizations, and deaths. For this study, Jefferson County’s CTT database was used to capture daily COVID-19 outcomes—i.e., cases, hospitalizations, and deaths. These outcomes included all reportable events among Jefferson County residents. Hospitalizations were recorded based on the patient’s residential ZIP code and were not limited to hospitals located within Jefferson County; admissions to out-of-county facilities were included if the patient resided in Jefferson County. Race was categorized as White (70.00%), Black (20.25%), or Other (9.75%). The “Other” category included individuals identifying as Asian (2.26%), American Indian or Alaska Native (0.93%), Native Hawaiian or Other Pacific Islander (0.17%), Multiracial (0.80%), or other races (5.59%). This recoding was necessary due to small sample sizes within these individual groups, which limited the practicality of conducting meaningful subgroup analyses.

#### 2.2.3. Study Data

The LMPHW event and CTT databases were merged by residential ZIP code and date for Jefferson County. The merged dataset was then aggregated by ZIP code and year-specific week or month to generate counts of COVID-19 vaccinations and outcomes. This process was repeated to create race-specific datasets. Descriptive statistics—mean and standard deviation (SD)—of vaccinations and outcomes are reported in Table 1. Additionally, total counts and population-based rates per 100,000 residents for vaccinations and outcomes during the study period are presented in the Appendix A.

### 2.3. Outcome Variables

COVID-19 outcomes—case, hospitalization, and death.

### 2.4. Independent Variable of Interest

The independent variable of interest was the number of COVID-19 vaccine doses administered through LMPHW-coordinated events, including standing sites, mobile units, and homebound mobile units.

### 2.5. Covariates

ZIP codes were included to account for observed and unobserved effects of ZIP code–specific non-time-varying factors, such as baseline demographics, socioeconomic status (SES), and neighborhood resources. Depending on the unit of time used in the analysis, dummy variables for each week or month by year were included to capture time-varying effects that are common across ZIP codes, such as seasonal trends, shifts in public health guidance, or changes in virus transmission. Finally, ZIP code-specific time dummies (ZIP code × week – or month–year interactions) were added to adjust for time-varying characteristics unique to each ZIP code, such as localized outbreaks or changes in ZIP code–level resources over time that could influence COVID-19 outcomes.

### 2.6. Statistical Model

Multiple linear regression models were used to estimate the association between the number of COVID-19 vaccine doses administered at LMPHW-coordinated events and each outcome—number of cases, hospitalizations, and deaths. Models were adjusted for the independent variable and covariates, with analyses conducted for the following time lags: one week, two weeks, three weeks, four weeks, and one month (Table 2). Fixed-effects regression models stratified by race (White, Black, and Other) were also conducted for each time lag to assess differences in associations across racial groups (Table 3).

Below is the model expression:
$$Y_{\mathrm{zt}}\;=\;\alpha \;+\;\beta {Vax}_{z,t-k}^{LMPHW}+\lambda_{z}+\mu_{t}+(\lambda_{z}\times \mu_{t})+\varepsilon_{\mathrm{zt}}$$where subscripts z and t indicate ZIP code and time, respectively. The variable Y_zt_ represents the COVID-19 outcome—number of cases, hospitalizations, or deaths—in ZIP code z in time (week or month) t. The variable ${Vax}_{z,t-k}^{LMPHW}$ is the number of COVID-19 vaccine doses administered at LMPHW-coordinated events in ZIP code z, k months (k = 1) or k weeks (k = 1, 2, 3, 4) before time t. The coefficient of this variable, β, is the coefficient of interest in this study. The variables λ and μ are ZIP code-specific and time-specific fixed-effects dummy variables, respectively. Factors associated with λ (e.g., population size, population racial and ethnic mix, and socioeconomic status) can be considered time invariant during the study period within the ZIP codes. The variable μ captures effects that potentially influence all ZIP codes at a specific time (e.g., weather conditions and the national or state public health policy environment). The term λ_z_ × μ_t_ represents the interaction between ZIP code and points in time, capturing time-specific factors associated with ZIP codes. The variable ε is the error term.

Adjusted coefficients for the independent variable are reported in Tables 2 and 3. Unadjusted coefficients are provided in the Appendix A. Model coefficients (β) are interpreted as the change in number of cases, hospitalizations, or deaths per 100 LMPHW-coordinated vaccine doses administered (β × 100). The critical value for statistical significance (*p*-value) was set at <0.05. Analyses were conducted using KNIME v5.1 and STATA 18.

To assess the robustness of the main models, several sensitivity analyses were conducted. First, to account for potential nonlinearity in the relationship between COVID-19 outcomes and vaccinations, Poisson and negative binomial regression models were estimated for each outcome—number of COVID-19 cases, hospitalizations, and deaths—using number of vaccine doses at each time lag (one week, two weeks, three weeks, four weeks, and one month), adjusting for the same covariates as the main models. Second, multiple linear regression models were estimated for each outcome, simultaneously including all weekly lagged vaccine dose variables and controlling for the same covariates. For each model in the sensitivity analysis, marginal effects were estimated because they allow for comparison with linear regression results. These marginal effects are presented in the Appendix A. Additionally, maps showing COVID-19 case, hospitalization, and death rates, as well as LMPHW-coordinated vaccine doses per 100,000 residents by ZIP code, are provided as supplementary figures to illustrate spatial patterns across Jefferson County Appendix A.

## 3. Results

From December 2020 to May 2022, trends in COVID-19 outcomes in Jefferson County, Kentucky, exhibited distinct temporal patterns. Monthly COVID-19 cases peaked most sharply between December 2021 and January 2022, with earlier surges observed around January 2021 and August 2021 (Figure 1).

Hospitalization trends followed a nearly identical pattern, with sharp peaks during each of these three periods (Figure 2). COVID-19-related deaths showed a slightly different trend, with sharp peaks in January 2021 and January 2022, but a more moderate increase in September 2021 (Figure 3). In contrast, the number of COVID-19 vaccine doses administered at LMPHW-coordinated events peaked between March and May 2021, and steadily declined thereafter (Figure 1, Figure 2 and Figure 3).

Spatial patterns of COVID-19 case, hospitalization, and death rates, as well as LMPHW-coordinated vaccine doses per 100,000 residents by ZIP code, highlight geographic variation in COVID-19 burden and vaccination reach across Jefferson County Appendix A. Overall, higher rates of COVID-19 cases, hospitalizations, and deaths were concentrated in several western and southwestern ZIP codes, whereas higher LMPHW-coordinated vaccine doses were observed in both high-burden areas and select central ZIP codes.

The overall mean (SD) number of COVID-19-positive cases in Jefferson County was 8246.06 (11,082.42) per month, 1927.65 (2767.30) per week, and 277.96 (434.00) per day. The corresponding mean (SD) for COVID-19-related hospitalizations was 255.83 (179.50) per month, 59.81 (48.20) per week, and 8.62 (7.56) per day. COVID-19-related deaths averaged 77.89 (60.02) per month, 18.21 (15.70) per week, and 2.63 (2.67) per day (Table 1).

**Table 1 vaccines-13-00901-t001:** Descriptive Statistics of COVID-19 Cases, Hospitalizations, Deaths, and LMPHW-Coordinated Vaccination Events–Doses among Residents of Jefferson County, Kentucky, December 2020–May 2022.

	Overall		White		Black		Other	
	Mean	^1^ SD	Mean	SD	Mean	SD	Mean	SD
Part 1A: COVID-19 Outcomes (Monthly)								
Reported COVID-19-Positive Cases	8246.06	11,082.42	6284.50	8640.60	1421.28	1811.49	540.28	648.40
COVID-19-Related Hospitalizations	255.83	179.50	190.00	138.20	50.44	32.69	15.39	11.05
COVID-19-Related Deaths	77.89	60.02	65.28	50.77	9.11	7.29	3.50	3.31
Part 1B: COVID-19 Outcomes (Weekly)								
Reported COVID-19-Positive Cases	1927.65	2767.30	1469.10	2126.59	332.25	487.52	126.30	168.10
COVID-19-Related Hospitalizations	59.81	48.20	44.42	37.29	11.79	8.97	3.60	3.49
COVID-19-Related Deaths	18.21	15.70	15.26	13.20	2.13	2.33	0.82	1.20
Part 1C: COVID-19 Outcomes (Daily)								
Reported COVID-19-Positive Cases	277.96	434.00	211.84	331.59	47.91	79.33	18.21	26.92
COVID-19-Related Hospitalizations	8.62	7.56	6.40	5.97	1.70	1.74	0.52	0.84
COVID-19-Related Deaths	2.63	2.67	2.20	2.33	0.31	0.61	0.12	0.35
Part 2A: Number of LMPHW Events								
Monthly	84.50	67.44	-	-	-	-	-	-
Weekly	19.75	16.32	-	-	-	-	-	-
Daily	2.85	4.16	-	-	-	-	-	-
Part 2B: Number of LMPHW Event Doses								
Monthly	795.78	825.87	-	-	-	-	-	-
Weekly	186.03	217.92	-	-	-	-	-	-
Daily	26.82	57.00	-	-	-	-	-	-

^1^ Standard Deviation; Louisville Metro Public Health and Wellness Department (LMPHW)-coordinated vaccine events: standing sites, mobile units, homebound.

Among White residents, mean (SD) COVID-19-positive cases were 6284.50 (8640.60) per month, 1469.10 (2126.59) per week, and 211.84 (331.59) per day. Hospitalizations averaged 190.00 (138.20) per month, 44.42 (37.29) per week, and 6.40 (5.97) per day. Mean (SD) deaths were 65.28 (50.77) per month, 15.26 (13.20) per week, and 2.20 (2.33) per day.

Among Black residents, mean (SD) COVID-19-positive cases were 1421.28 (1811.49) per month, 332.25 (487.52) per week, and 47.91 (79.33) per day. Hospitalizations averaged 50.44 (32.69) per month, 11.79 (8.97) per week, and 1.70 (1.74) per day. Mean (SD) deaths were 9.11 (7.29) per month, 2.13 (2.33) per week, and 0.31 (0.61) per day.

Among residents of other racial groups, mean (SD) COVID-19-positive cases were 540.28 (648.40) per month, 126.30 (168.10) per week, and 18.21 (26.92) per day. Hospitalizations averaged 15.39 (11.05) per month, 3.60 (3.49) per week, and 0.52 (0.84) per day. Deaths were 3.50 (3.31) per month, 0.82 (1.20) per week, and 0.12 (0.35) per day.

The mean (SD) number of LMPHW-coordinated vaccine events was 84.50 (67.44) per month, 19.75 (16.32) per week, and 2.85 (4.16) per day. These events administered an average of 795.78 (825.87) doses per month, 186.03 (217.92) per week, and 26.82 (57.00) per day.

In the adjusted multiple linear regression models, LMPHW-coordinated vaccine doses were significantly associated with reductions in COVID-19-related deaths (Table 2). Specifically, every 100 doses administered was associated with a decrease of 0.3 deaths at a one-week lag (β = −0.003, SE = 0.001); 0.2 deaths at a two-week lag (β = −0.002, SE = 0.001); and 0.4 deaths at a one-month lag (β = −0.004, SE = 0.002). However, these effect sizes were minimal and nearly identical across models. No significant association was found at the three-week and four-week lags.

A small but statistically significant association was also observed between vaccine doses and COVID-19-related hospitalizations in the one-week lag model. Every 100 LMPHW-coordinated doses administered was associated with a reduction of 0.4 hospitalizations at a one-week lag (β = −0.004, SE = 0.002). No significant associations were observed for hospitalizations or positive cases at any other time lag.

In the fixed-effects regression models for White residents, every 100 LMPHW-coordinated vaccine doses administered was associated with modest reductions in COVID-19-related deaths across time lags: 0.2 deaths at a one-week lag (β = −0.002, SE = 0.001); 0.2 deaths at a two-week lag (β = −0.002, SE = 0.001); 0.1 deaths at a four-week lag (β = −0.001, SE = 0.001); and 0.3 deaths at a one-month lag (β = −0.003, SE = 0.001) (Table 3). Significant reductions in hospitalizations were also observed at the one-week and three-week lags. In both models, every 100 vaccine doses administered was associated with an identical decrease of 0.3 hospitalizations (β = −0.003, SE = 0.001). No significant associations were observed between vaccine doses and COVID-19-positive cases at any time lag among White residents.

In the fixed-effects regression models for Black residents, no significant associations were found between vaccine doses coordinated by LMPHW and any COVID-19 outcomes across all time lags. Among residents of other racial groups, vaccine doses were not significantly associated with COVID-19-related deaths at any time lag. However, every 100 doses administered was associated with a small but statistically significant and identical reduction of 0.1 hospitalizations at both the two-week and four-week lags (β = −0.001, SE = 0.001).

In the fixed-effects regression models for residents of other racial groups, almost all of the time-lag models showed significant negative associations between COVID-19-positive cases and vaccine doses coordinated by LMPHW. Every 100 doses administered was associated with reductions of 1.5 cases at a one-week lag (β = −0.015, SE = 0.005); 1.6 cases at a two-week lag (β = −0.016, SE = 0.005); and 1.3 cases at a three-week lag (β = −0.013, SE = 0.005). However, the one-month lag model produced a larger reduction of 3.4 cases per 100 doses (β = −0.034, SE = 0.015), suggesting a stronger delayed association between vaccine administration and reductions in COVID-19 cases among residents of other racial groups.

The results from the supplementary models were largely consistent with the findings from the main multiple linear regressions (Appendix A). In the Poisson regression models, statistically significant negative associations were observed between vaccine doses and COVID-19-related deaths across all time lags, with the strongest effects at one-week and two-week lags (Appendix A). For instance, every 100 LMPHW-coordinated vaccine doses administered was associated with small reductions in COVID-19-related deaths across time lags: 0.3 deaths at a one-week lag (β = −0.003, SE = 0.001); 0.2 deaths at a two-week lag (β = −0.002, SE = 0.001); and 0.3 deaths at a one-month lag (β = −0.003, SE = 0.001). These patterns closely matched the results of the main model (Table 2), where significant reductions in deaths were also observed at one-week, two-week, and one-month lags.

The negative binomial models showed similar results to the multiple linear regressions, particularly for deaths, reinforcing the robustness of the mortality findings Appendix A. Every 100 LMPHW-coordinated vaccine doses administered was associated with small reductions in COVID-19-related deaths across time lags: 0.3 deaths at a one-week lag (β = −0.003, SE = 0.001); 0.2 deaths at a two-week lag (β = −0.002, SE = 0.001); and 0.3 deaths at a one-month lag (β = −0.003, SE = 0.001). For hospitalizations, the one-week lag effect observed in the main model was directionally consistent but not statistically significant in the Poisson and negative binomial models. For COVID-19-positive cases, both Poisson and negative binomial models revealed statistically significant reductions at all time lags, while the main model yielded negative but non-significant associations.

In multiple linear regression models that included all lagged vaccine dose variables simultaneously, significant reductions in deaths were again observed at the one-week (β = −0.002, SE = 0.001) and two-week lags (β = −0.001, SE = 0.001) Appendix A. Associations with cases and hospitalizations in these models remained directionally consistent but were reduced and not statistically significant, particularly for hospitalizations. Taken together, these sensitivity analyses support the robustness of the main findings, especially for COVID-19-related deaths.

## 4. Discussion

This study estimated the ZIP code-level effect of COVID-19 vaccine doses coordinated by the LMPHW on COVID-19 outcomes—cases, hospitalizations, and deaths—stratified by race in Jefferson County, Kentucky. The findings demonstrate modest but statistically significant reductions in COVID-19 cases, hospitalizations, and deaths associated with vaccine doses, particularly at one week post-vaccination. Additionally, notable variation in vaccine–outcome associations by race was observed.

Although the observed effects of vaccine doses on COVID-19 outcomes were modest, these findings align with prior observational studies showing that incremental increases in vaccine coverage can meaningfully reduce population-level cases, hospitalizations, and deaths [15,21,22]. Moreover, the association between COVID-19 vaccine doses and reductions in outcomes—which was particularly evident in the one-week lag model—supports existing evidence of a rapid immune response following vaccination [6].

Vaccine protection against infection and severe outcomes (i.e., hospitalization and death) wanes over time [6]. This trend is reflected in the temporal patterns observed in Jefferson County, where a decline in COVID-19 outcomes coincided with a peak in LMPHW-coordinated vaccine doses administered from March to May 2021 (Figure 1, Figure 2 and Figure 3).

This period aligned with Kentucky’s COVID-19 vaccine prioritization policy, which followed CDC ACIP guidelines. Phase 1c, initiated on 1 March 2021, prioritized adults aged 60 and older and essential workers. By May 6 (Phase 2), eligibility expanded to adults aged 40 and older, and by May 13 (Phase 3), to all individuals aged 16 and older [14].

The post-peak decline in vaccine administration after mid-2021 likely reflects reduced demand following high initial uptake among eligible populations, while the resurgence in COVID-19 cases during late 2021 and early 2022 coincided with the emergence of the Omicron variant, which showed reduced protection from prior infection and vaccination [7]. Together, the regression results and observed temporal trends underline the importance of maintaining up-to-date vaccination coverage in reducing population-level burden—particularly during periods of high transmission and the emergence of new variants.

The findings from the fixed-effects models emphasize the importance of vaccine equity in improving population health outcomes and reducing infection risk. Among White residents, vaccine doses were significantly associated with reductions in COVID-19-related hospitalizations and deaths across multiple time lags. While significant associations were also observed for COVID-19-positive cases and hospitalizations among residents of other racial groups, no significant associations were observed for deaths. In contrast, none of the vaccine–outcome associations were statistically significant among Black residents.

One potential explanation for this pattern is that vaccine coverage remained lower overall in Black communities despite LMPHW outreach. Previous studies in Jefferson County found that Black residents had significantly lower vaccine uptake than White residents across all age groups [12,13,14]. LMPHW-coordinated doses may have represented a larger share of total vaccine access for Black residents, while for White residents, they were probably supplemented with higher coverage from other sources (i.e., hospitals, pharmacies, and non-LMPHW vaccine events). As a result, the population-level impact of LMPHW doses may have been harder to detect among Black residents due to limited overall vaccine coverage.

In addition, systemic barriers such as historical mistrust of public health institutions, disparities in access to healthcare, and disproportionate occupational or housing-related exposure risks may have further contributed to persistent disparities in COVID-19 outcomes, even in the presence of localized vaccination efforts [23,24].

These disparities may be shaped by differences in healthcare access, vaccine confidence, exposure risk, and the prevalence of underlying health conditions across racial groups [25,26,27,28]. While this study does not establish causality, the observed racial variation in vaccine–outcome associations underlines the need for community-based vaccination campaigns that promote equitable vaccine uptake by improving access, fostering acceptance, and building trust in marginalized communities. These findings may assist public health policy and emergency preparedness efforts aimed at reducing racial disparities through equity-driven vaccination strategies in future outbreak responses.

## 5. Limitations

This study has several strengths, including the use of ZIP code-level data, accuracy of COVID-19-related hospitalizations and deaths data, and race-stratified analyses. However, several limitations exist. First, this was an ecological analysis conducted at the ZIP code level; therefore, causal inferences cannot be drawn. In particular, an assessment of individual-level associations between vaccination and COVID-19 outcomes was not possible.

Second, the vaccine data reflected only doses coordinated at LMPHW events and did not capture vaccines administered at events organized by other entities in the county. Given the realistic assumption that vaccination at other organizations’ events is associated with a decrease in both COVID-19 outcomes in the county and vaccination at LMPHW events, the provided estimates undervalue the true effect of vaccination at LMPHW events on COVID-19 outcomes in the county. Moreover, the proportion of total doses administered by LMPHW likely varied across ZIP codes and racial groups, which may have influenced the observed associations. Therefore, the presented estimates should be considered as lower bounds of the true effects of the LMPHW vaccination program. Additionally, individuals vaccinated through LMPHW services may differ from those vaccinated through other service providers in terms of health status, socioeconomic factors, or COVID-19 risk. Because LMPHW events—particularly mobile units and homebound outreach—were designed to reach underserved populations, this may introduce selection bias into the observed associations.

Third, the study relied on reported COVID-19-positive cases, which may be subject to underreporting due to under-testing. Third, due to differences in demographics, vaccine implementation strategies, and healthcare access in Jefferson County, the results may not be generalizable to other regions.

Fourth, our analysis did not distinguish between first and second doses or account for the 14-day period typically required to achieve full clinical protection following completion of the primary series [29]. Therefore, the associations identified in this study should be interpreted as reflecting directional effects of cumulative vaccine dose delivery on population-level outcomes, rather than definitive evidence of full vaccine-induced immunity.

Fifth, our analysis used absolute counts of vaccine doses and outcomes rather than population-adjusted rates. This approach makes it possible to express the dose–response relationship in absolute terms, which is useful, since a public program’s impact would generally be measured in total number of people affected (e.g., as part of a cost-effectiveness analysis). However, in using absolute counts, our analysis does not quantify the typical ecological association, i.e., the association between the incremental vaccination rate achieved by the LMPHW campaign (the dose) and COVID-19-related incidence rates (the response).

Sixth, in the sensitivity analysis, models that include multiple lags may be influenced by multicollinearity and unmeasured confounding among the lagged predictors. Seventh, stratifying models by race reduced the number of observations within each group, which may have limited our ability to detect smaller effects—especially for less frequent outcomes like hospitalizations and deaths.

Lastly, we observed that COVID-19 cases, hospitalizations, and deaths were proportionally lower among Black and Other racial groups compared to their population share. This may have been influenced by differences in age structure, healthcare access, underreporting, or testing availability, which were not fully accounted for in the analysis. Because the outcomes were not age-adjusted and were based on raw counts, these differences should be interpreted with caution.

## 6. Conclusions

COVID-19 vaccine doses coordinated by the LMPHW were associated with modest reductions in COVID-19 cases, hospitalizations, and deaths in Jefferson County, Kentucky. These findings emphasize the importance of the timely uptake of recommended vaccines to protect against emerging variants. Additionally, racial variation observed in vaccine–outcome associations highlights the need to address disparities through equitable, community-based vaccine strategies and access to care. Such efforts are vital to improving population health outcomes and strengthening preparedness for future public health emergencies.

## Figures and Tables

**Figure 1 vaccines-13-00901-f001:**
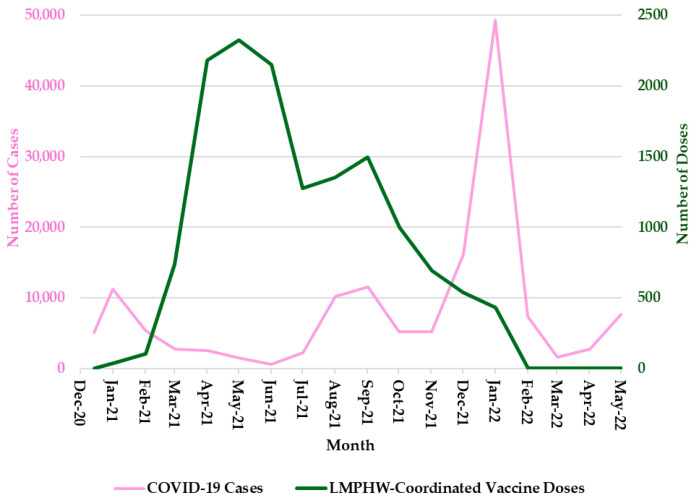
Monthly COVID-19 Cases and LMPHW-Coordinated Vaccine Doses in Jefferson County, Kentucky. The left-hand vertical axis shows the total number of COVID-19 cases per month. The right-hand vertical axis displays the total number of vaccine doses administered each month through LMPHW-coordinated events. Y-axis scales differ across panels to preserve visibility of temporal patterns for each outcome. Figures are intended to illustrate within-outcome trends, not direct comparisons across outcomes.

**Figure 2 vaccines-13-00901-f002:**
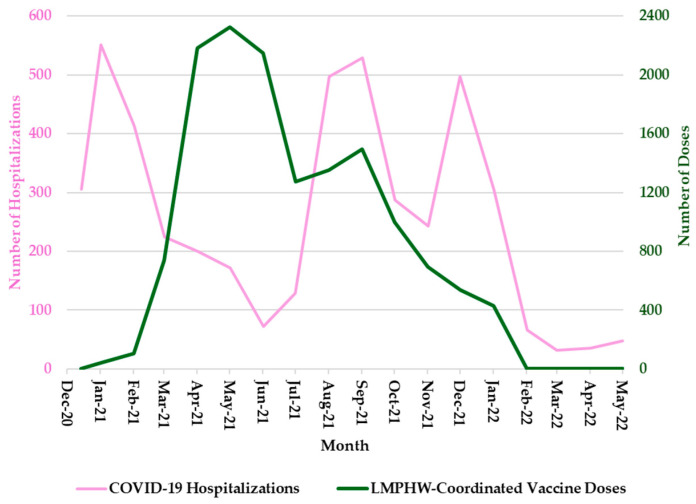
Monthly COVID-19 Hospitalizations and LMPHW-Coordinated Vaccine Doses in Jefferson County, Kentucky. The left-hand vertical axis shows the total number of COVID-19-related hospitalizations per month. The right-hand vertical axis displays the total number of vaccine doses administered each month through LMPHW-coordinated events. Y-axis scales differ across panels to preserve visibility of temporal patterns for each outcome. Figures are intended to illustrate within-outcome trends, not direct comparisons across outcomes.

**Figure 3 vaccines-13-00901-f003:**
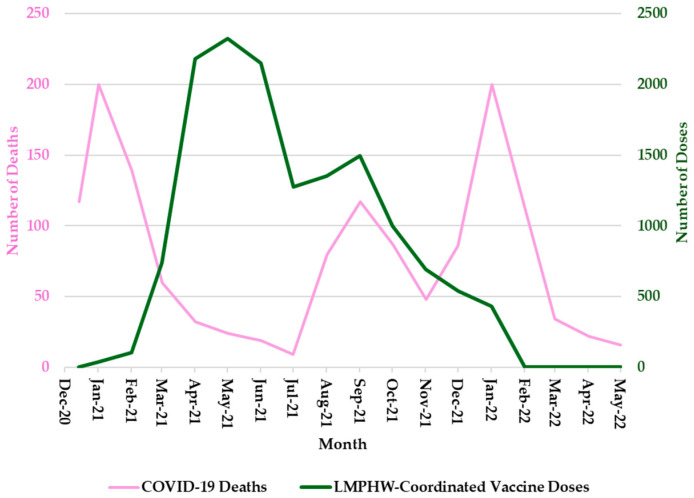
Monthly COVID-19 Deaths and LMPHW-Coordinated Vaccine Doses in Jefferson County, Kentucky. The left-hand vertical axis shows the total number of COVID-19-related deaths per month. The right-hand vertical axis displays the total number of vaccine doses administered each month through LMPHW-coordinated events. Y-axis scales differ across panels to preserve visibility of temporal patterns for each outcome. Figures are intended to illustrate within-outcome trends, not direct comparisons across outcomes.

**Table 2 vaccines-13-00901-t002:** Associations between the Number of COVID-19 Vaccine Doses Administered at the LMPHW-Coordinated Events and COVID-19 Outcomes in Jefferson County, Kentucky (Adjusted Regression Models).

	One-Week Lag	Two-Week Lag	Three-Week Lag	Four-Week Lag	One-Month Lag
^1^ COVID-19-Related Deaths					
^2^ Number of Doses	−0.003 ***	−0.002 ***	−0.001	−0.001	−0.004 **
	(0.001)	(0.001)	(0.001)	(0.001)	(0.002)
^1^ COVID-19-Related Hospitalizations					
^2^ Number of Doses	−0.004 **	0.000	−0.002	0.001	−0.003
	(0.002)	(0.002)	(0.002)	(0.002)	(0.004)
^1^ COVID-19 Positive Cases					
^2^ Number of Doses	−0.116	−0.103	−0.076	−0.043	−0.200
	(0.069)	(0.070)	(0.070)	(0.070)	(0.230)

^1^ Dependent variables; ^2^ COVID-19 vaccine doses coordinated by the Louisville Metro Public Health and Wellness Department (LMPHW); Standard errors in parentheses; *** *p* < 0.01, ** *p* < 0.05.

**Table 3 vaccines-13-00901-t003:** Associations between the Number of COVID-19 Vaccine Doses Administered at the LMPHW-Coordinated Events and COVID-19 Outcomes in Jefferson County, Kentucky (Race-Specific Adjusted Fixed-Effects Regression Models).

	One-Week Lag	Two-Week Lag	Three-Week Lag	Four-Week Lag	One-Month Lag
**White**					
^1^ COVID-19-related Deaths					
^2^ Number of Doses	−0.002 ***	−0.002 ***	−0.001	−0.001 **	−0.003 **
	(0.001)	(0.001)	(0.001)	(0.001)	(0.001)
^1^ COVID-19-related Hospitalizations					
^2^ Number of Doses	−0.003 ***	−0.000	−0.003 **	0.001	−0.004
	(0.001)	(0.001)	(0.001)	(0.001)	(0.003)
^1^ COVID-19 Positive Cases					
^2^ Number of Doses	−0.082	−0.074	−0.052	−0.031	−0.138
	(0.054)	(0.054)	(0.054)	(0.054)	(0.181)
**Black**					
^1^ COVID-19-related Deaths					
^2^ Number of Doses	−0.000	−0.000	−0.000	−0.000	−0.000
	(0.000)	(0.000)	(0.000)	(0.000)	(0.000)
^1^ COVID-19-related Hospitalizations					
^2^ Number of Doses	−0.000	0.001	0.001	0.000	0.001
	(0.001)	(0.001)	(0.001)	(0.000)	(0.001)
^1^ COVID-19 Positive Cases					
^2^ Number of Doses	−0.019	−0.013	−0.010	−0.002	−0.028
	(0.015)	(0.015)	(0.015)	(0.015)	(0.044)
**Other**					
^1^ COVID-19-related Deaths					
^2^ Number of Doses	−0.000	0.000	0.000	−0.000	−0.000
	(0.000)	(0.000)	(0.000)	(0.000)	(0.000)
^1^ COVID-19-related Hospitalizations					
^2^ Number of Doses	−0.000	−0.001 **	0.000	−0.001 ***	−0.001
	(0.000)	(0.000)	(0.000)	(0.000)	(0.000)
^1^ COVID-19 Positive Cases					
^2^ Number of Doses	−0.015 ***	−0.016 ***	−0.013 ***	−0.010	−0.034 **
	(0.005)	(0.005)	(0.005)	(0.005)	(0.015)

^1^ Dependent variables; ^2^ COVID-19 vaccine doses coordinated by the Louisville Metro Public Health and Wellness Department (LMPHW); Standard errors in parentheses; *** *p* < 0.01, ** *p* < 0.05.

## Data Availability

The datasets analyzed in this study cannot be publicly accessible.

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
