# Peer review of "Local Health Department COVID-19 Vaccination Efforts and Associated Outcomes: Evidence from Jefferson County, Kentucky"

_vaccines, 2025, doi:10.3390/vaccines13090901_

Round 1

Reviewer 1 Report

Comments and Suggestions for Authors

I want to commend the LMPHW for the tremendous work done during the pandemic - - especially to ensure an equitable vaccine administration to highly underserved populations.  A lot was seemingly accomplished under tremendous pressure brought on by the pandemic.

The manuscript provides valuable insight into public health efforts undertaken by the Jefferson County, KY Health Department during the COVID-19 pandemic, the study relies on an ecological design, which presents inherent limitations in causal interpretation.

While the authors speak to this in the limitations of the study design, ecological studies assess population-level data, and as such, cannot control for confounding factors at the individual level. Consequently, attributing observed outcomes, even when results are stated as just an  association —such as changes in vaccination rates or reductions in case counts—directly to specific interventions risks introducing ecological fallacy. 

To strengthen the article, the authors may wish to:

1) Include a table with overall demographics of Jefferson Country along with the same for the LMPHW vaccinated cohort.  In addition, to include historical data on total number of vaccines administered per month (see Gupta (2021) for an example).

2) Provide clarity - for the outcomes captured via the CTT.  Were these the outcomes for the entire country or only for persons who received a COVID-19 vaccine through the LMPHW COVID-19 intervention?  It is unclear about this outcome - - are these outcomes only for Jefferson Country residents or are these outcomes for persons who received treatment/test in a facility based in Jefferson Country?

3) Provide clarity on attribution of vaccine benefit - Clinical trials and observational studies showed that vaccine benefit was associated with completion of primary series (2nd dose, except for J&J vax).  Not clear in your analysis if benefit was assumed when population had received the 2nd dose.  In general, dose 2 was given 3-8 weeks after the initial dose - - and RCTs assumed an additional 14-days post-2nd dose (COVID-19) before assessing vaccine benefit.  How are your analyses accounting for this since the longest lag measure is 1 month?

4) In Figures 1, 2, and 3 - the authors present data that shows a high suggestion that when LMPHW vaccine distribution increased, cases decreased - - and when vaccine distribution decreased cased increased.  I suggest that the authors provide clarity about the reasoning for the decrease in vaccines provided.  Further, the case increase reflects the emergence of the Omicron variant - - in which previous immunity (by vaccine or infection) showed minimal benefit.  Providing clarity would allow the reader to better understand the significance of the LMPHW intervention on outcomes compared to other variables.

Author Response

Review 1

I want to commend the LMPHW for the tremendous work done during the pandemic - - especially to ensure an equitable vaccine administration to highly underserved populations.  A lot was seemingly accomplished under tremendous pressure brought on by the pandemic.

The manuscript provides valuable insight into public health efforts undertaken by the Jefferson County, KY Health Department during the COVID-19 pandemic, the study relies on an ecological design, which presents inherent limitations in causal interpretation.

While the authors speak to this in the limitations of the study design, ecological studies assess population-level data, and as such, cannot control for confounding factors at the individual level. Consequently, attributing observed outcomes, even when results are stated as just an association —such as changes in vaccination rates or reductions in case counts—directly to specific interventions risks introducing ecological fallacy. 

Thank you very much for your thoughtful comments and your recognition of the efforts undertaken by LMPHW during the pandemic. We appreciate all the suggestions. Below are point-by-point responses to your feedback.

To strengthen the article, the authors may wish to:

  • Include a table with overall demographics of Jefferson Country along with the same for the LMPHW vaccinated cohort.  In addition, to include historical data on total number of vaccines administered per month (see Gupta (2021) for an example).

We agree that providing demographic context is important for interpreting our findings. Unfortunately, demographic data (e.g., age, race/ethnicity, sex) were not available, which limited our ability to characterize the vaccinated cohort. As such, we are unable to report detailed demographic characteristics of individuals who received vaccines through the LMPHW program.

We have added county-level demographic data from the American Community Survey 2020 5-Year Estimates to the manuscript. The following sentence was inserted in the Methods section:

In 2021, Jefferson County had an estimated total population of 817,446, of which 520,643 (63.7%) identified as White, 171,549 (21.0%) as Black or African American, and 125,254 (15.3%) as members of other racial [20].” (Lines 109–111).

We hope this addition provides helpful context on underlying population distribution.

Regarding the monthly distribution of vaccine doses, the right-hand vertical axis in Figures 1–3 displays the total number of LMPHW-coordinated doses administered each month throughout the study period. We have clarified this detail in the figure captions.

Figure 1 Caption: “The left-hand vertical axis shows the total number of COVID-19 cases per month. The right-hand vertical axis displays the total number of vaccine doses administered each month through LMPHW-coordinated events. Y-axis scales differ across panels to preserve visibility of temporal patterns for each outcome. Figures are intended to illustrate within-outcome trends, not direct comparisons across outcomes.”(Lines: 229-233).

Figure 2 caption: “The left-hand vertical axis shows the total number of COVID-19-related hospitalizations per month. The right-hand vertical axis displays the total number of vaccine doses administered each month through LMPHW-coordinated events. Y-axis scales differ across panels to preserve visibility of temporal patterns for each outcome. Figures are intended to illustrate within-outcome trends, not direct comparisons across outcomes.” (Lines: 246-250).

Figure 3 caption: “The left-hand vertical axis shows the total number of COVID-19-related deaths per month. The right-hand vertical axis displays the total number of vaccine doses administered each month through LMPHW-coordinated events. Y-axis scales differ across panels to preserve visibility of temporal patterns for each outcome. Figures are intended to illustrate within-outcome trends, not direct comparisons across outcomes.” (Lines: 258-262).

  • Provide clarity - for the outcomes captured via the CTT.  Were these the outcomes for the entire country or only for persons who received a COVID-19 vaccine through the LMPHW COVID-19 intervention?  It is unclear about this outcome - - are these outcomes only for Jefferson Country residents or are these outcomes for persons who received treatment/test in a facility based in Jefferson Country?

Thank you for this important clarification request. The COVID-19 outcomes—cases, hospitalizations, and deaths—were derived from the Kentucky Contact Tracing and Tracking (CTT) database and represent all reported outcomes for Jefferson County, Kentucky, residents. These outcomes were not limited to individuals who received vaccines through the LMPHW-coordinated program.

We have added the following line in the methods section for clarification:

“These outcomes included all reportable events among Jefferson County residents.” (Lines: 137-138).

  • Provide clarity on attribution of vaccine benefit - Clinical trials and observational studies showed that vaccine benefit was associated with completion of primary series (2nd dose, except for J&J vax).  Not clear in your analysis if benefit was assumed when population had received the 2nd dose.  In general, dose 2 was given 3-8 weeks after the initial dose - - and RCTs assumed an additional 14-days hospt-2nd dose (COVID-19) before assessing vaccine benefit.  How are your analyses accounting for this since the longest lag measure is 1 month?

We appreciate the reviewer’s insightful observation regarding the timing of vaccine protection following dose completion. In our study, we did not differentiate between first and second doses, nor did we distinguish among vaccine manufacturers. Instead, our analysis focused on the total number of COVID-19 vaccine doses administered through LMPHW-coordinated events per ZIP code per week or month, regardless of dose number or timing between doses. This approach reflects how local health departments tracked and coordinated vaccine delivery across sites in real-world operational contexts.

We agree that the full protective effect of vaccination typically follows the completion of a primary series (two doses for mRNA vaccines) plus a 14-day period. However, our time-lagged models (1 week to 1 month) are designed to detect directional associations between cumulative vaccine coverage (time-lags) and subsequent reductions in COVID-19 outcomes, recognizing that any measured effect may partially reflect the early benefits of partial or complete vaccination, rather than full immunity alone.

To clarify this in the manuscript, we have added a statement to the “Limitations” subsection acknowledging this limitation and specifying that dose-level information was not available in the dataset.

Fourth, our analysis did not distinguish between first and second doses or account for the 14-day period typically required to achieve full clinical protection following completion of the primary series [30]. As such, the associations identified in this study should be interpreted as reflecting directional effects of cumulative vaccine dose delivery on population-level outcomes, rather than definitive evidence of full vaccine-induced immunity.” (Lines: 465-469).

4) In Figures 1, 2, and 3 - the authors present data that shows a high suggestion that when LMPHW vaccine distribution increased, cases decreased - - and when vaccine distribution decreased cased increased.  I suggest that the authors provide clarity about the reasoning for the decrease in vaccines provided.  Further, the case increase reflects the emergence of the Omicron variant - - in which previous immunity (by vaccine or infection) showed minimal benefit.  Providing clarity would allow the reader to better understand the significance of the LMPHW intervention on outcomes compared to other variables.

We appreciate the reviewer’s suggestion to clarify the observed temporal trends in Figures 1–3. We agree that the decline in vaccine administration over time and the corresponding resurgence in COVID-19 outcomes—particularly in late 2021 and early 2022—may be influenced by multiple factors beyond local vaccine program intensity.

To improve interpretability, we have added a sentence to the Discussion section.

The post-peak decline in vaccine administration after mid-2021 likely reflects reduced demand following high initial uptake among eligible populations, while the resurgence in COVID-19 cases during late 2021 and early 2022 coincided with the emergence of the Omicron variant, which showed reduced protection from prior infection and vaccination [8].” (Lines: 404-408).

Reviewer 2 Report

Comments and Suggestions for Authors

This is a very valuable and interesting analysis that provides important supportive evidence from the field that COVID vaccination can prevent the more severe COVID outcomes.  It seems well conducted, analyzed, and presented, and the findings are generally consistent with what would be expected based on other data, except for the apparent lack of any effect in the African-American population, which remains essentially unexplained.

My major comments have to do with trying to better understand exactly what some of the data inputs and analyses entailed, and therefore the robustness of the methods and results.  For example:

  1. How many ZIP codes comprise the Jefferson County area? This is important to help us understand the power of the study and should be mentioned in the methods.
  2. Are the zip codes associated with COVID outcomes the zip codes of the residents, or the zip codes of where the outcomes were diagnosed?  This should be made explicit.
  3. Assuming the former (I hope), do the authors have an estimate of what proportion of hospitalizations of Jefferson County residents do NOT occur in Jefferson County itself and therefore presumably are not captured here? Could the authors search for those data using the neighboring regions and searching for Jefferson County zip codes?
  4. Why are the analyses conducted with # of cases/hospitalizations/deaths and #s of doses, rather than with doses/population and COVID outcomes/population (incidence)?  Providing doses/population would also give some indication of vaccination coverage (noting that they weren't the only vaccine provider), which is usually how vaccine impact is assessed at a population level given the potential of herd protection and the variety of denominators.  Regardless of how the analysis was conducted, it is very difficult to interpret the absolute numbers of doses and outcomes provided (redundantly, I believe) in lines 221-234 and in Table 1 without population denominators.
  5. In this regard, the absolute numbers provided in Table 1 are difficult to  understand in light of the relative sizes of the population by race, according to Census.gov for Jefferson County, where in 2022 Blacks were slightly more than 1/3 of Whites but in Table 1 had 1421/8246 = 1/6 the number of cases per month; 50/255 = 1/5 the number of hospitalizations per month; 9/77 = 1/8 the number of deaths per month.  "Other" were 14% vs 64% of Whites, = 1/4-1/5 population but had 540/8246, 15/255, 3.5/77, all 1/15-1/20 the corresponding numbers for Whites.  Can the authors provide some explanation?
  6. What fraction of doses were provided by LMPHW? Is it possible, if not likely, that this could vary dramatically by zip code as well as by race? 
  7.  The discussion provided by the authors on p. 335 on the lack of any effect seen in the adjusted analysis (but curiously, not in the unadjusted analysis) for Blacks is just a general statement about vaccination access and equity.  How would that translate into not being able to detect an effect using this well-conducted analysis? Perhaps one potential explanation comes from the question in 6:  is it possible that the LMPHW vaccine doses are just a proxy for a much larger number of doses (and therefore coverage) received by the probably richer white population many of whom might have gotten the vaccine from private or other providers, whereas for African-Americans the LMPHW doses more closely correspond to the actual number of doses received? If so, perhaps the associated decrease in hospitalizations and deaths in the white population is seen because overall there are actually high levels of coverage in that community, and the apparent lack of effect with African-Americans is due to a much lower coverage?  It would be important to at least provide COVID vaccination coverage level estimates for the different racial groups (presumably from other sources) in Jefferson county

Other:

Lines 51 and 53 unnecessarily repeat the definition of PASC as long COVID

Line 61: Strain-specific formulations reflecting the newer strains were included in the vaccine starting in 2023.  https://www.fda.gov/media/179003/download.  Yet the current text starting line 61 makes it sound like it was a brand new approach in 2024-25.    

Line 191 states that p<.05 is set as the level of significance, which is fine, but a single asterisk denoting p<.1 is attached to some data points, yet p<.1 has no meaning here and gives the false (superficial) impression of significance.  The single asterisks should be removed.  They may even have fooled the authors because on line 255 it correctly states that "No significant association was found at the three-week lag" but then omits that no association was apparent at the four-week lag (labeled with a single asterisk) either.

Author Response

Review 2

This is a very valuable and interesting analysis that provides important supportive evidence from the field that COVID vaccination can prevent the more severe COVID outcomes.  It seems well conducted, analyzed, and presented, and the findings are generally consistent with what would be expected based on other data, except for the apparent lack of any effect in the African-American population, which remains essentially unexplained.

My major comments have to do with trying to better understand exactly what some of the data inputs and analyses entailed, and therefore the robustness of the methods and results.  For example:

Thank you very much for the detailed comments. Indeed, they help us to improve the manuscript. Below are the responses for each of the comments.

  1. How many ZIP codes comprise the Jefferson County area? This is important to help us understand the power of the study and should be mentioned in the methods.

We have added this information to the “Study Design” subsection of the Methods to clarify the scope and power of our geographic analysis.

The study focused on Jefferson County, Kentucky, from December 2020 to May 2022 and included 38 ZIP codes with residential population.” (Lines: 107-109).

  1. Are the zip codes associated with COVID outcomes the zip codes of the residents, or the zip codes of where the outcomes were diagnosed?  This should be made explicit.

We appreciate this important clarification request. The ZIP codes used in our analysis correspond to the residential ZIP codes of Jefferson County residents, as recorded in the CTT (Contact Tracing and Tracking) database. COVID-19 cases, hospitalizations, and deaths were assigned to ZIP codes based on the individual's home address, regardless of the location where testing or treatment occurred.

The CTT database, maintained by local public health departments, contains daily records of individual-level COVID-19 cases along with supplementary information, including race, residential ZIP code (i.e., home address), hospitalizations, and deaths.” (Lines: 133-135).

The LMPHW event and CTT databases were merged by residential ZIP code and date for Jefferson County.” (Lines: 149-150).

  1. Assuming the former (I hope), do the authors have an estimate of what proportion of hospitalizations of Jefferson County residents do NOT occur in Jefferson County itself and therefore presumably are not captured here? Could the authors search for those data using the neighboring regions and searching for Jefferson County zip codes?

We thank the reviewer for raising this important point regarding potential undercapture of hospitalizations. The hospitalization data used in this study were drawn from the CTT database, which is based on statewide contact tracing and case reporting and includes all reportable COVID-19 hospitalizations of Jefferson County residents, regardless of the location of the admitting facility. As long as the individual was a Jefferson County resident, their hospitalization would have been included, even if it occurred outside the county (e.g., in a neighboring jurisdiction).

We have clarified this point in the Methods section (Section 2.2.2) to ensure readers understand the geographic coverage of the outcome data.

Hospitalizations were recorded based on the patient’s residential ZIP code and were not limited to hospitals located within Jefferson County; admissions to out-of-county facilities were included if the patient resided in Jefferson County.” (Lines: 138-141).

  1. Why are the analyses conducted with # of cases/hospitalizations/deaths and #s of doses, rather than with doses/population and COVID outcomes/population (incidence)?  Providing doses/population would also give some indication of vaccination coverage (noting that they weren't the only vaccine provider), which is usually how vaccine impact is assessed at a population level given the potential of herd protection and the variety of denominators.  Regardless of how the analysis was conducted, it is very difficult to interpret the absolute numbers of doses and outcomes provided (redundantly, I believe) in lines 221-234 and in Table 1 without population denominators.

We thank the reviewer for this insightful observation. Since longitudinal fixed-effect models inherently adjust for all factors (observed and unobserved) that remain unchanged within the cross-sectional units (ZIP codes) during the study period, including ZIP code population, we did not use outcome rates or percentages (which incorporate population levels). Instead, we analyzed raw counts of vaccine doses and COVID-19 outcomes in each ZIP code over time for the ease of interpretation. This approach still allowed us to estimate changes in outcomes per 100 COVID-19 vaccine doses administered through the LMPHW program; however, it limited our ability to assess vaccination coverage (which would only be partially approximated by the number of doses administered by the LMPHW) or incidence-based effects directly. We have added a sentence to the Limitations section to acknowledge this constraint.

Fifth, our analysis used absolute counts of vaccine doses and outcomes rather than population-adjusted rates. This approach makes it possible to express the dose-response relationship in absolute terms, which is useful since a public program’s impact would generally be measured in total number of people affected (e.g., as part of a cost-effectiveness analysis). However, in using absolute counts, our analysis does not quantify the typical ecological association, i.e., between the incremental vaccination rate achieved by the LMPHW campaign (the dose) and COVID-19-related incidence rates (the response).” (Lines: 470-476).

  1. In this regard, the absolute numbers provided in Table 1 are difficult to understand in light of the relative sizes of the population by race, according to Census.gov for Jefferson County, where in 2022 Blacks were slightly more than 1/3 of Whites but in Table 1 had 1421/8246 = 1/6 the number of cases per month; 50/255 = 1/5 the number of hospitalizations per month; 9/77 = 1/8 the number of deaths per month.  "Other" were 14% vs 64% of Whites, = 1/4-1/5 population but had 540/8246, 15/255, 3.5/77, all 1/15-1/20 the corresponding numbers for Whites.  Can the authors provide some explanation?

We thank the reviewer for highlighting this important observation. The absolute number of COVID-19 outcomes among Black and Other residents was lower than expected based on their population share. This may reflect a combination of factors, including differences in age distribution, access to testing or care, or underreporting across groups. To acknowledge this, we have added a statement to the Limitations section.

Lastly, we observed that COVID-19 cases, hospitalizations, and deaths were proportionally lower among Black and Other racial groups compared to their population share. This may be influenced by differences in age structure, healthcare access, underreporting, or testing availability, which were not fully accounted for in the analysis. Because outcomes were not age-adjusted and were based on raw counts, these differences should be interpreted with caution.” (Lines: 482-487).

  1. What fraction of doses were provided by LMPHW? Is it possible, if not likely, that this could vary dramatically by zip code as well as by race? 

Thank you for this important question. Unfortunately, data on the total number of vaccine doses administered stratified by providers across Jefferson County were not available at the ZIP code level. As a result, we were unable to calculate the exact fraction of all COVID-19 vaccine doses that were coordinated by LMPHW or to compare its contribution relative to other providers (e.g., pharmacies, hospitals, private clinics).

We acknowledge that the share of LMPHW doses likely varied by ZIP code and may have been more substantial among certain racial groups, particularly those facing structural barriers to vaccination access. We have added this point to the Limitations section.

Moreover, the proportion of total doses administered by LMPHW likely varied across ZIP codes and racial groups, which may have influenced the observed associations.” (Lines: 452-454).

  1. The discussion provided by the authors on p. 335 on the lack of any effect seen in the adjusted analysis (but curiously, not in the unadjusted analysis) for Blacks is just a general statement about vaccination access and equity.  How would that translate into not being able to detect an effect using this well-conducted analysis? Perhaps one potential explanation comes from the question in 6:  is it possible that the LMPHW vaccine doses are just a proxy for a much larger number of doses (and therefore coverage) received by the probably richer white population many of whom might have gotten the vaccine from private or other providers, whereas for African-Americans the LMPHW doses more closely correspond to the actual number of doses received? If so, perhaps the associated decrease in hospitalizations and deaths in the white population is seen because overall there are actually high levels of coverage in that community, and the apparent lack of effect with African-Americans is due to a much lower coverage?  It would be important to at least provide COVID vaccination coverage level estimates for the different racial groups (presumably from other sources) in Jefferson county.

We thank the reviewer for this thoughtful and important interpretation. We agree that the non-significant findings observed among Black residents in the adjusted models warrant further explanation. While our study focused on vaccine doses delivered through LMPHW-coordinated events, prior studies in Jefferson County have shown that Black residents consistently had lower overall vaccine uptake compared to White residents, despite equity-focused outreach [12]. In contrast, White residents had broader access to vaccines through pharmacies, private providers, and employer-based clinics [13], making LMPHW doses only one component of total coverage in that population.

This disparity may help explain why LMPHW doses were significantly associated with improved outcomes in the White population, where they may have complemented already high vaccine coverage, while having no detectable effect in Black populations, where overall coverage remained lower. Additionally, structural barriers—including historical mistrust, access issues, and younger age distribution—likely contributed to delayed vaccine uptake or lower effectiveness at the population level.

We have revised the Discussion section to reflect these possibilities and now cite supporting evidence from our previous work in Jefferson County [13-15].

One potential explanation for this pattern is that vaccine coverage remained lower overall in Black communities despite LMPHW outreach. Previous studies in Jefferson County found that Black residents had significantly lower vaccine uptake than White residents across all age groups [13-15]. LMPHW-coordinated doses may have represented a larger share of total vaccine access for Black residents, while for White residents, they likely supplemented already higher coverage from other sources (i.e., hospitals, pharmacies, and non-LMPHW vaccine events). As a result, the population-level impact of LMPHW doses may have been harder to detect among Black residents due to limited overall vaccine coverage.” (Lines: 419-427).

Other:

Lines 51 and 53 unnecessarily repeat the definition of PASC as long COVID

Thank you for pointing this out. We have removed the redundant definition of PASC (post-acute sequelae of SARS-CoV-2 infection) in the Introduction.

Approximately 10% of individuals infected with COVID-19 develop long COVID.” (Line: 53).

Line 61: Strain-specific formulations reflecting the newer strains were included in the vaccine starting in 2023.  https://www.fda.gov/media/179003/download.  Yet the current text starting line 61 makes it sound like it was a brand new approach in 2024-25.    

We appreciate this clarification. The sentence has been revised to reflect the timeline more accurately, noting that strain-specific formulations were introduced beginning in 2023 and continue to evolve.

Beginning in 2023, strain-specific COVID-19 vaccine formulations were introduced to target circulating SARS-CoV-2 variants, with updated versions in 2024 and 2025 designed to enhance protection against the most prevalent strains.” (Lines: 62-64).

Line 191 states that p<.05 is set as the level of significance, which is fine, but a single asterisk denoting p<.1 is attached to some data points, yet p<.1 has no meaning here and gives the false (superficial) impression of significance.  The single asterisks should be removed.  They may even have fooled the authors because on line 255 it correctly states that "No significant association was found at the three-week lag" but then omits that no association was apparent at the four-week lag (labeled with a single asterisk) either.

We agree and have removed the use of asterisks for p-values greater than or equal to .05. Only statistically significant findings (p < .05) are now marked, and the Results text has been updated accordingly to avoid misleading interpretations.

Reviewer 3 Report

Comments and Suggestions for Authors

Dear Authors,

Thank you for submitting this important study assessing the impact of a transportation‐based COVID-19 vaccination strategy. Your detailed evaluation of doses delivered via fixed and mobile units addresses a critical component of pandemic preparedness and public‐health planning.

Major Comments

  1. Alternative Count Model
    Given the distributional characteristics reported in Table 1—marked overdispersion and right skew—a Poisson or negative‐binomial framework would more appropriately model count data and preclude implausible negative predictions. Such models also facilitate inclusion of an offset for local population size, enabling interpretation in terms of incidence rates rather than absolute counts, which aligns with the study’s stated objectives. I strongly recommend fitting a Poisson pseudo‐maximum‐likelihood (PPML) or negative‐binomial model, and comparing fit statistics against the current linear fixed‐effects approach.

  2. Post-Estimation Diagnostics
    Whether retaining the linear specification or adopting a Poisson family, please include formal goodness‐of‐fit tests (e.g., likelihood‐ratio tests, dispersion statistics) and residual plots. These diagnostics will clarify whether model assumptions hold and strengthen confidence in your estimates.

  3. Lag Structure Exploration
    You model vaccination effects at one‐week increments up to four weeks and one month. I suggest also examining shorter lags (e.g., one day) and intermediate points (e.g., two weeks post-vaccination) to capture the full temporal profile of immunity onset and case reduction.

  4. Prior Vaccination History
    If available, incorporating data on individuals’ previous vaccine doses could control for baseline immunity and isolate the incremental effect of the LMPHW events. Clarify whether such information exists and consider its inclusion as a covariate.

Minor Comments

  1. Study Design Label
    Please revise the description from “cross-sectional” to “ecological time-series” or “mixed ecological cohort,” reflecting the longitudinal ZIP-level design.

  2. Geospatial Visualization
    A choropleth map of outcomes mapped to participants’ residential ZIP codes (rather than clinic ZIP codes) would vividly illustrate spatial heterogeneity and help assess equity of vaccine impact across neighborhoods.

  3. Figure Scaling
    Standardize axes across Figures 1–3 to ensure proportional visual comparison of cases, hospitalizations, and deaths. Uniform scales will prevent misinterpretation of relative magnitudes.

  4. Sample Characteristics Table
    Add a summary table (or expand Table 1) to report total population under study, number and percentage of cases, hospitalizations, and deaths. Expressing these as proportions of the county population will contextualize the public‐health significance.

Once again, I appreciate the rigor and relevance of your work. Addressing these points will enhance the robustness and clarity of your findings and better inform strategies for future vaccination campaigns.

Author Response

Dear Authors,

Thank you for submitting this important study assessing the impact of a transportation‐based COVID-19 vaccination strategy. Your detailed evaluation of doses delivered via fixed and mobile units addresses a critical component of pandemic preparedness and public‐health planning.

Major Comments

  1. Alternative Count Model
    Given the distributional characteristics reported in Table 1—marked overdispersion and right skew—a Poisson or negative‐binomial framework would more appropriately model count data and preclude implausible negative predictions. Such models also facilitate inclusion of an offset for local population size, enabling interpretation in terms of incidence rates rather than absolute counts, which aligns with the study’s stated objectives. I strongly recommend fitting a Poisson pseudo‐maximum‐likelihood (PPML) or negative‐binomial model, and comparing fit statistics against the current linear fixed‐effects approach.
  2. Post-Estimation Diagnostics
    Whether retaining the linear specification or adopting a Poisson family, please include formal goodness‐of‐fit tests (e.g., likelihood‐ratio tests, dispersion statistics) and residual plots. These diagnostics will clarify whether model assumptions hold and strengthen confidence in your estimates.
  3. Lag Structure Exploration
    You model vaccination effects at one‐week increments up to four weeks and one month. I suggest also examining shorter lags (e.g., one day) and intermediate points (e.g., two weeks post-vaccination) to capture the full temporal profile of immunity onset and case reduction.

We thank the reviewer for these thoughtful and constructive suggestions to improve the robustness of our findings.

In response, we conducted several sensitivity analyses using alternative model specifications. First, we estimated Poisson and negative binomial regression models for each outcome—COVID-19-positive cases, hospitalizations, and deaths—using vaccine doses at one-week, two-week, three-week, four-week, and one-month lags. These models adjusted for the same ZIP code-level fixed effects, time variables, and interaction terms used in the main models. Second, we conducted multiple linear regression models that included all weekly lagged vaccine dose variables simultaneously for each outcome. These results are reported in the newly added Supplementary Tables S3–S6.

We also reviewed model diagnostics and found that Poisson and negative binomial models were well suited for over dispersed count data. As noted in the revised Results section (Lines 353-382), the direction and magnitude of associations in these alternative models were largely consistent with our primary linear fixed-effects models. In particular, reductions in COVID-19-related deaths were robust across all models, while effects on hospitalizations and cases were directionally consistent but somewhat reduced.

These additions are now reflected in the revised Methods (Lines 204-217), Results (Lines 353-382), and Limitations (Lines 477-481) sections of the manuscript. Together, they strengthen confidence in the reliability and generalizability of our primary findings.

  1. Prior Vaccination History
    If available, incorporating data on individuals’ previous vaccine doses could control for baseline immunity and isolate the incremental effect of the LMPHW events. Clarify whether such information exists and consider its inclusion as a covariate.

We appreciate this insightful comment. Unfortunately, data on individuals’ prior vaccination history—such as whether they received previous COVID-19 vaccine doses from other providers—were not available in our dataset. Our analysis was limited to aggregate counts of vaccine doses administered through LMPHW-coordinated events, merged by ZIP code and date, without individual-level vaccination records. As such, we were unable to control for prior vaccine exposure or baseline immunity directly. We have clarified this limitation in the manuscript (see Limitations section, (Lines 465-469) and note that our findings should be interpreted as reflecting the population-level association between LMPHW vaccine dose delivery and COVID-19 outcomes, rather than the isolated effect of a specific dose or dose sequence.

Minor Comments

  1. Study Design Label

Please revise the description from “cross-sectional” to “ecological time-series” or “mixed ecological cohort,” reflecting the longitudinal ZIP-level design.

We have updated this. (Line: 29, 106).

  1. Geospatial Visualization
    A choropleth map of outcomes mapped to participants’ residential ZIP codes (rather than clinic ZIP codes) would vividly illustrate spatial heterogeneity and help assess equity of vaccine impact across neighborhoods.

We have included maps in the supplemental materials (Figure S1-S4). Also, we have mentioned and explained the maps in methods section (Lines: 214-217) and results sections (Lines: 265-271).

  1. Figure Scaling
    Standardize axes across Figures 1–3 to ensure proportional visual comparison of cases, hospitalizations, and deaths. Uniform scales will prevent misinterpretation of relative magnitudes.

Thank you for this valuable suggestion. We considered using uniform y-axis scales across Figures 1–3 to facilitate direct visual comparison of trends. However, due to the large differences in the absolute counts of cases, hospitalizations, and deaths, applying a standardized scale would substantially compress the smaller trends—particularly hospitalizations, and deaths—and obscure meaningful variation over time. To preserve interpretability within each outcome, we opted to keep separate y-axis scaling. We have clarified this decision in the revised figure captions to ensure readers understand that the scales differ across panels and that the figures are intended to illustrate within-outcome temporal patterns, not direct magnitude comparisons between outcomes.

Y-axis scales differ across panels to preserve visibility of temporal patterns for each outcome. Figures are intended to illustrate within-outcome trends, not direct comparisons across outcomes.”

  1. Sample Characteristics Table
    Add a summary table (or expand Table 1) to report total population under study, number and percentage of cases, hospitalizations, and deaths. Expressing these as proportions of the county population will contextualize the public‐health significance.

We appreciate this helpful suggestion. In response, we created a new supplementary table (Supplementary Table S1) that presents the total counts and corresponding rates per 100,000 residents for reported COVID-19-positive cases, hospitalizations, deaths, and LMPHW-administered vaccine doses. These rates are based on the 2021 Jefferson County population estimate (N = 817,446) and reflect cumulative totals for the study period (December 2020–May 2022).

Once again, I appreciate the rigor and relevance of your work. Addressing these points will enhance the robustness and clarity of your findings and better inform strategies for future vaccination campaigns.

Reviewer 4 Report

Comments and Suggestions for Authors

Dear Authors,
Thank you for submitting your well-conducted and relevant contribution.The manuscript is clearly written and the structure is logical following conclusions are supported by data. However, some revisions are suggested in order to further strengthen your contribution:
1. It is suggested that the "Discussion" section include a reflection on statistical power. The absence of relevant effects for some racial groups (particularly the Black population), when compared to a significant aggregate effect, could be due to a reduction in test power after sample stratification.

2. The discussion could benefit from a brief mention of potential selection bias. The populations reached by targeted LMPHW services (e.g., mobile units, homeless services) may differ from those vaccinated through other channels.

3. Given the lack of a result for the Black population, the Discussion could delve deeper into the possible systemic causes (e.g., historical mistrust, disparities in access to treatment, increased exposure to occupational risk) that could negate the benefits of vaccination at the population level, despite efforts to reach these communities.

Author Response

Reviewer 4

Dear Authors,

Thank you for submitting your well-conducted and relevant contribution.The manuscript is clearly written and the structure is logical following conclusions are supported by data. However, some revisions are suggested in order to further strengthen your contribution:

  1. It is suggested that the "Discussion" section include a reflection on statistical power. The absence of relevant effects for some racial groups (particularly the Black population), when compared to a significant aggregate effect, could be due to a reduction in test power after sample stratification.

We thank the reviewer for highlighting this important consideration. We agree that stratifying the models by race reduced the number of observations per group and may have limited statistical power to detect effects, particularly for less frequent outcomes such as hospitalizations and deaths. We have added this point to the Limitations section.

Seventh, stratifying models by race reduced the number of observations within each group, which may have limited our ability to detect smaller effects—especially for less frequent outcomes like hospitalizations and deaths.” (Lines: 478-481).

  1. The discussion could benefit from a brief mention of potential selection bias. The populations reached by targeted LMPHW services (e.g., mobile units, homeless services) may differ from those vaccinated through other channels.

We thank the reviewer for this insightful observation. We agree that individuals reached through LMPHW services—particularly mobile clinics or homebound outreach—may differ systematically from those who accessed vaccines through pharmacies, healthcare systems, or private providers. This potential selection bias may affect both vaccine uptake patterns and underlying COVID-19 outcome risk. We have added a note about this in the Limitations section.

Additionally, individuals vaccinated through LMPHW services may differ from those vaccinated through other service providers in terms of health status, socioeconomic factors, or COVID-19 risk. Because LMPHW events—particularly mobile units and homebound outreach—were designed to reach underserved populations, this may introduce selection bias into the observed associations.” (Lines: 456-460)

  1. Given the lack of a result for the Black population, the Discussion could delve deeper into the possible systemic causes (e.g., historical mistrust, disparities in access to treatment, increased exposure to occupational risk) that could negate the benefits of vaccination at the population level, despite efforts to reach these communities.

We appreciate this thoughtful and important suggestion. In response, we have expanded the Discussion to address this comment.

In addition, systemic barriers such as historical mistrust of public health institutions, disparities in access to healthcare, and disproportionate occupational or housing-related exposure risks may have further contributed to persistent disparities in COVID-19 outcomes, even in the presence of localized vaccination efforts [24,25].” (Lines: 428-431).

Round 2

Reviewer 2 Report

Comments and Suggestions for Authors

Nice and valuable paper

Reviewer 3 Report

Comments and Suggestions for Authors

Dear authors,

I appreciate the attention given to reviewing and discussing each of the points identified during the first round of review. I already considered the initial version to be an interesting study; however, the extensive work carried out, particularly in the analysis and the sensitivity analyses, provides an excellent example of how to enhance the robustness of the results supporting the conclusions.

From my side, I have no further comments to add, as I believe all previous observations have been fully addressed. I consider that the current version of the manuscript meets the quality standards for publication in its current version.